# SUT: Active Defects Probing for Transcompiler Models

**Mengnan Qi[1,*]  Yufan Huang[1,*]  Maoquan Wang[1,*]  Yongqiang Yao[1,*]**
**Zihan Liu[2,*]  Bin Gu[3,4]  Colin Clement[1]  Neel Sundaresan[1]**
[1] Microsoft Cloud and AI
[2] Shanghai Jiao Tong University
[3] School of Artificial Intelligence, Jilin University
[4] Mohamed bin Zayed University of Artificial Intelligence
{mengnanqi,yufanhuang,maoquanwang,yongqiangyao}@microsoft.com
lzh123@sjtu.edu.cn

## Abstract

Program translation, i.e. transcompilation has been attracting increasing attention from researchers due to its enormous application value. However, we observe that current program translating models still make elementary syntax errors, particularly when the source language uses syntax elements not present in the target language, which is exactly what developers are concerned about while may not be well exposed by frequently used metrics such as BLEU, CodeBLEU and Computation Accuracy. In this paper, we focus on evaluating the model's ability to address these basic syntax errors and developed an novel active defects probing suite, the Syntactic Unit Tests (SUT) and highly interpretable evaluation harness including Syntax Unit Test Accuracy (SUT Acc) metric and Syntax Element Test Score (SETS), to help diagnose and promote progress in this area. Our Syntactic Unit Test fills the gap in the community for a fine-grained evaluation dataset for program translation. Experimental analysis shows that our evaluation harness is more accurate, reliable, and in line with human judgments compared to previous metrics.

## 1 Introduction

Program translation or transcompilation aims to automatically convert source code from one programming language (e.g., C++) into another (e.g., Java), meanwhile the results should preserve the program function and ideally follow the target language conventions. The recent proposal of transcompilation datasets such as CodexGLUE(Lu et al., 2021) benchmark, GeeksforGeeks dataset for computation accuracy(Rozière et al., 2020) and XLCoST(Zhu et al., 2022) benchmark has propelled related research in the program translation domain (Feng et al., 2020; Kanade et al., 2020; Ahmad et al., 2021a; Guo et al., 2021; Lachaux et al.,

2021; Clement et al., 2020; Rozière et al., 2020, 2021). These innovations have resulted in impressive improvements in neural transcompilation, but most models still make elementary syntax errors, especially when the source and target languages have different sets of syntactic structures.

Existing datasets are often criticized for their lack of granularity, which can mask deficiencies in basic syntactic elements that humans care. Their data are derived from randomly sampled code stores or programming contest websites, and the code they contain may not even be executable. As a result, researchers have had to rely on sequence similarity metrics like BLEU, ROUGE, or CodeBLEU to evaluate the quality of code translation. However, these metrics have been shown to be inadequate for objectively assessing the quality of code translation.

In this paper, we propose Syntactic Unit Tests(SUT), which leverage basic syntax structures of each language along with unit tests to verify the correctness of the translations. Table1 presents a comparative analysis of SUT against other code translation evaluation datasets. Our experiments show that SUT performance of state-of-the-art transcompilers like ChatGPT has definite room for improvement, demonstrating the value of the metric for judging future model improvements. Further, we propose the Syntax Unit Test Accuracy (SUT Acc) and Syntax Element Test Score (SETS), using statistical inference to analysis individual syntax element performance, which is more in line with translation quality that humans concern about. In summary, we make the following contributions:

- We build SUT, a new set of unit tests for programming languages C++, Java, Python, and C#, interpretable syntactic categories for each, and a test harness environment for compiling and executing these unit tests given hypothesis translations from any model, which fills the gap in the field of a fine-grained evaluation

---

Table 1: Comparison against other code translation eval datasets (Py - Python, JS - JavaScript).

| Datasets | Support Languages | Code Form | Avg. tokens | Executable | Limited by Parallel Data | Syntax Element Analysis |
|---|---|---|---|---|---|---|
| CodeXGLUE | Java, C# | methods | 43.1 | $\times$ | $\checkmark$ | $\times$ |
| XLCoST | C++, Java, Py, C#, JS, PHP, C | code snippets | 22.83 | $\times$ | $\checkmark$ | $\times$ |
| AVATAR | Java, Python | programs | 147.9 | $\checkmark$ | $\checkmark$ | $\times$ |
| TransCoder GeeksforGeeks | Java, Python, C++ | programs | 111.9 | $\checkmark$ | $\checkmark$ | $\times$ |
| SUT | Java, Python, C++, C# | unit tests | 28.15 | $\checkmark$ | $\times$ | $\checkmark$ |

dataset for program translation.

- We propose a more accurate reliable, and in line with human judgments evaluation harness, including SUT Acc and SETS, which focus on analysing model performance on fine-grained elements.

- We evaluate a variety of the most advanced program translation models in a unified manner. Our experiments reveal syntactic element errors in which these models exhibit deficiencies, and help researchers working in this field to get potential avenues for improvement.

## 2 Related Work

### 2.1 Program Translation Datasets

Both CodexGLUE (Lu et al., 2021) and XLCoST (Zhu et al., 2022) benchmark provide complete parallel pair functions for program translation tasks, while XLCoST also provides random code snippet level data pairs. However, a major issue occurs that they do not account for the environmental dependencies of running the program. Transcoder (Rozière et al., 2020) extracts code from the GeeksforGeeks website and provides a test environment, then AVATAR (Ahmad et al., 2021b) collects more parallel datasets for Java-Python translation. However, the generated code must be completely correct to pass the test, while developers often want to clearly locate which part of the code generates errors to facilitate quick debugging.

### 2.2 Metric for Program Translation

For evaluation of Transcompilation, early studies followed the NLP analogy and mainly employed BLEU (Papineni et al., 2002) to evaluate the generated translations. But the n-gram overlap does not directly reflect the quality of the resulting translations. CodeBLEU (Ren et al., 2020) improves upon BLEU by using features from the abstract syntax tree (AST) and data flow of variables. Nevertheless, these static metrics still cannot tell whether the translated code has the same behavior as the

source code. TransCoder (Rozière et al., 2020) test whether the translated program outputs the same results as the source program for each test case. They defined Computational Accuracy (CA@$K$) to evaluate the probability that a given program can be translated and produce the correct output in $K$ attempts. The results of the CA meet human expectations, but it is too expensive to build a test environment for code with complex dependencies.

## 3 Syntactic Unit Tests Suite

### 3.1 Syntactic Unit Tests (SUT)

Inspired by how humans learn programming, we note that it is essential for a program translator to master the syntax elements to learn how to write programs in a language. To evaluate whether the translation model has a solid understanding of language characteristics, tests designed for each syntax element are necessary. Similar to Computational Accuracy by the authors of TransCoder (Rozière et al., 2020), we introduce scenario programs and unit tests which must be satisfied for each syntax elements of the languages under consideration. The idea is that, if the model fully understands a syntax element, it should translate the test function correctly and thus pass the unit tests. Further, failures are explicitly associated with syntax elements, allowing us to interpret the performance of the model and plan for improvements which target the most impactful elements.

The program in each test case accepts one or more parameters as input and generates the expected output. Each case from SUT has four properties: 1) the name and category of the syntax element 2) the source program 3) the input of the function 4) the expected output of the function. We manually craft SUT datasets for four languages: C++, Java, Python, and C#, following programming language instruction sets and show some specific cases in Appendix A.

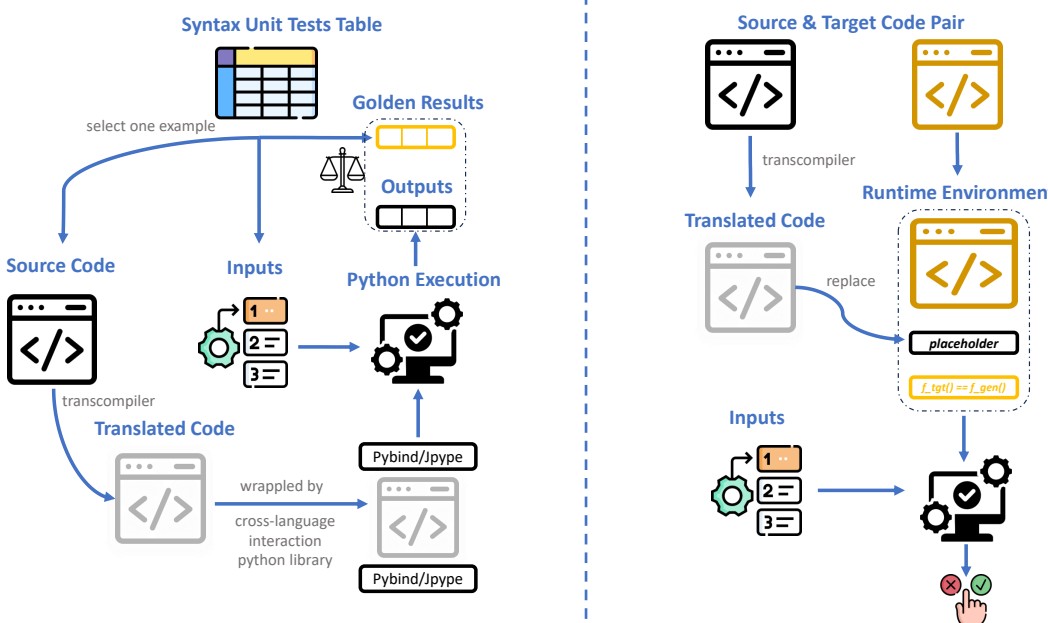

Figure 1: **Comparison the construct process between SUT Acc(the left) and Computation Accuracy(the right).** Our method wraps the translated target language code with a cross-language interaction python library, and executes it uniformly in the python environment. We only compare the final execution results, so the target language does not need to provide the corresponding golden code.

## 3.2 Evaluation Harness

### 3.2.1 Syntax Unit Test Accuracy

Transcoder (Rozière et al., 2020) randomly selected coding problems from the GeeksforGeeks website and used Computation Accuracy as an evaluation metric. Our syntax unit test accuracy metric evaluates the model's pass rate on disentangled test samples just containing basic elements. Figure 1 compares the calculation process of Our SUT Acc and Transcoder's CA. Our method does not rely on parallel data pairs, which not only reduces potential data noise, but also can be expanded upon with relative ease. The translated function is wrapped in a Python unittest[1] function. Leveraging language interoperation tools such as JPype[2] and pybind11[3], we can use Python to compile and execute the programs in the other languages under consideration. Combined with the above unit tests, the system injects each input of the input list to the unit test function and compares the output with the expected output, defining it as passing if they match. Syntax Unit Test Accuracy calculates pass rate of execution results and measures the various basic syntactic correctness of the model.

### 3.2.2 Syntax Element Test Score (SETS)

Based on the results of SUT Acc, we can get some sense of translation performance on the syntax elements. However, sometimes SUT case contains multiple syntax elements as some syntax elements are not by themselves able to render a valid program. Therefore, we further disentangle the statistical effects of multiple syntax elements in each SUT case. For each SUT test case $i$, we use the parsing tool Tree-sitter[4] to obtain the set of syntax elements, and construct a matrix $A_{ij}$ which is 1 if that test program contains syntax element $j$. Then when we get the results $y_i$ of each unit test, which may empirically be 1 or 0 if the test passed or failed. We are interested in the fail rate of syntax element $j$ though, and so we model the relationship as $y = Ax$, where $y_i = \log P(\text{fail test}_i)$ is the empirical log fail rate of test $i$, $x_j = \log P(\text{fail syntax element}_j)$ is the unknown fail rate of syntax element $j$, and $A$ is the known relationship between test $i$ and syntax element $j$. We then use Lasso regression to get consistent results for the estimated log fail rate of each syntax element. The higher the score, the worse the model is at translating such basic elements.

---

[1] https://docs.python.org/3/library/unittest.html

[2] https://jpype.readthedocs.io/

[3] https://pybind11.readthedocs.io/

[4] https://tree-sitter.github.io/tree-sitter/

Table 2: Comparison the translation performance of these program translators on our SUT and test dataset in Transcoder. "To X" & "From X": average performance when translating to & from language X. The SUT Acc line shows the pass rate on our SUT. The CA line and the CA filter line respectively show the pass rate on the test dataset in Transcoder GeeksforGeeks and after the data errors in it are filterd.

| Model | EvalSet | To C++ | From C++ | To Ja | From Ja | To Py | From Py | To C# | From C# | Avg. |
|---|---|---|---|---|---|---|---|---|---|---|
| TransCoder | SUT Acc | 35.94% | 45.94% | 34.74% | 42.58% | 45.16% | 27.32% | – | – | **38.61%** |
| | CA | 56.20% | 56.10% | 50.85% | 64.40% | 48.05% | 34.60% | – | – | 51.70% |
| | CA filter | 61.04% | 60.85% | 55.08% | 69.96% | 52.22% | 37.53% | – | – | 56.11% |
| TransCoder-ST | SUT Acc | 42.02% | 49.59% | 43.67% | 57.42% | 53.28% | 31.95% | – | – | **46.32%** |
| | CA | 70.65% | 64.65% | 63.10% | 76.75% | 65.10% | 57.45% | – | – | 66.28% |
| | CA filter | 76.73% | 70.13% | 68.35% | 83.38% | 70.74% | 62.31% | – | – | 71.94% |
| text-davinci-003 | SUT Acc | 55.99% | 65.31% | 58.89% | 63.54% | 65.60% | 41.05% | 58.08% | 68.67% | **59.64%** |
| | CA | 77.84% | 74.72% | 72.09% | 79.79% | 74.14% | 69.57% | – | – | 74.69% |
| | CA filter | 84.54% | 81.06% | 78.09% | 86.82% | 80.71% | 75.45% | – | – | 81.11% |
| gpt-3.5-turbo | SUT Acc | 60.62% | 70.45% | 62.50% | 67.18% | 78.99% | 50.31% | 63.53% | 77.70% | **66.41%** |
| | CA | 88.55% | 85.21% | 80.60% | 88.71% | 85.67% | 80.89% | – | – | 84.94% |
| | CA filter | 96.17% | 92.45% | 87.31% | 98.61% | 93.09% | 87.74% | – | – | 92.56% |

## 4 Experimental Results

### 4.1 Evaluation of Program Translators

Table 2 shows the translation performance of influential program translators with our SUT and the unit test proposed in TransCoder (Rozière et al., 2020). For a detailed introduction to these models, please refer to Appendix B. The test data in TransCoder still remains some data impurity(mainly on the input of the test sample and the mismatch of the given translation pair logic). Table 2 demonstrates that after these errors were addressed, certain translation tasks almostly no longer confused the SOTA models, while the SUT remained challenging for them. It is worth noting that although most test functions are elementary, models still fail to translate in many cases. Our test results indicate a crucial bottleneck in most existing program translators: these models focus on achieving program mappings in a macroscopic manner, without specially designed tasks to learn code snippet mappings associated with each individual syntax. This may limit the further improvement of model performance.

### 4.2 Performance on Syntactic Element

In this section, we present the translation performance of our models on the base elements using SETS scores. For the convenience display at fig 2, it is summarized into 5 categories, Variable & Constant, Datatype, Data Structure, Operator, and Loop & Branch. More detailed analysis can be seen in Appendix C. The diagnose report can assist re-

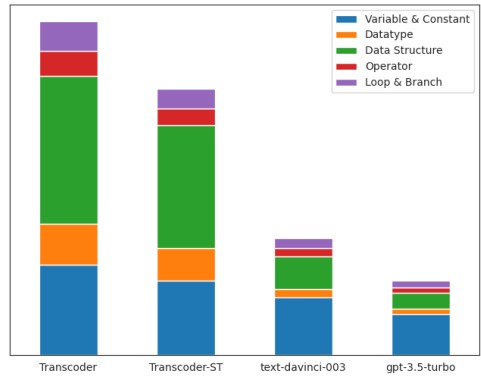

Figure 2: **Error Distribution of These Tested Models.**

searchers in identifying the model's weak areas and addressing these issues to quickly improve the model's performance. In the appendix D, we provide a brief demonstration of how our reports can be utilized to enhance model performance.

## 5 Conclusion

In this paper, we introduce an interpretable benchmark via Syntactic Unit Test and associated Syntax Element Test Score, which can diagnose the specific weakness of a given translation model. It fills the gap in the field of a fine-grained evaluation dataset for program translation. By our designed evaluation suites, researchers can actively probe deficiencies in their models about specific basic syntactic elements and improve their performance in a targeted manner.

## Limitations

While our work provides an evaluation tool for exploring code translation models on fine-grained underlying semantic elements. However, limited by time and manpower, our data set is still not comprehensive enough, and currently only supports four languages: Python, Java, C++ and C#. This article only uses the S-Tuning experiment (shown in the appendix D) to show the prospect of fine-grained data sets for improving the model effect. Regarding how to use these diagnostic information, more effective solutions need to be proposed by the follow-up community.

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

# A  Examples of SUT Cases

## 1) C++

| name | src | input | expected_output |
| --- | --- | --- | --- |
| division operator | ```int foo(int a, int b) {```
```    int c = a / b;```
```    return c;```
```}``` | [ (9, 2), (9, 3), (9, 4) ] | [ 4, 3, 2 ] |
| conditional operator | ```int foo(int x, int y) {```
```    int z = (x > y) ? x : y;```
```    return z```
```}``` | [ (1, 2), (2, 1), (1, 1), (-1, -2), (-2, -1), (0, 0) ] | [ 2, 2, 1, -1, -1, 0 ] |
| break in for statement | ```int foo(int x, int y) {```
```    int s = 0;```
```    for(int i = 1; i <= x; ++i) {```
```        if (s > y) break;```
```        s += i;```
```    }```
```    return s;```
```}``` | [ (3, 1), (3, 2), (3, 3) ] | [ 3, 3, 6 ] |
| array index | ```int foo(int x) {```
```    int a[3];```
```    a[0] = 0;```
```    a[1] = 1;```
```    a[2] = 2;```

```    return a[x-1];```
```}``` | [ 1, 2, 3 ] | [ 0, 1, 2 ] |

Figure 3: Examples of C++ SUT cases.

## 2) Java

| name | src | input | expected_output |
| --- | --- | --- | --- |
| string length | ```int foo(String s) {```
```    return s.length;```
```}``` | [ "abcde" ] | [ 5 ] |
| if-else if-else condition | ```int foo(int a, int b) {```
```    if (a > b) {```
```        return 1;```
```    } else if (a == b) {```
```        return 0;```
```    } else {```
```        return -1;```
```    }```
```}``` | [ (4, 4) ] | [ 0 ] |
| do-while statement | ```int foo(int a) {```
```    int i = a;```
```    do {```
```        i++;```
```    }```
```    while (i < a);```
```    return i;```
```}``` | [ 3 ] | [ 4 ] |
| for each statement | ```int foo(int a) {```
```    int[] n = {1,2,3,4};```
```    for (int i : n) {```
```        a += i;```
```    }```
```    return a```
```}``` | [ 8 ] | [ 18 ] |

Figure 4: Examples of Java SUT cases.

## 3) Python

| name | src | input | expected_output |
|------|-----|-------|-----------------|
| raw string | ```python
def foo():
    a = r'\n'
    return a
``` | [ ] | [ "\\n" ] |
| conditional expression | ```python
def foo(a, b):
    return b if a > b else a
``` | [ (4, 9) ] | [ 4 ] |
| continue in for statement | ```python
def foo(x):
    a = list(range(10))
    s = 0
    for i in a:
        if i % x == 0:
            continue
        s += i
    return s
``` | [ 2, 3 ] | [ 25, 27 ] |
| stack top | ```python
def foo():
    s = []
    s.append(1)
    s.append(2)
    s.append(3)
    return s[-1]
``` | [ ] | [ 3 ] |

Figure 5: Examples of Python SUT cases.

## 4) C#

| name | src | input | expected_output |
|------|-----|-------|-----------------|
| double to string | ```csharp
string foo(double a) {
    return a.ToString();
}
``` | [ 3.14, -1.5 ] | [ "3.14", "-1.5" ] |
| switch statement | ```csharp
int foo(int a) {
    int b;
    switch (a) {
        case 1:
            b = a + 1;
            break;
        case 2:
            b = a + 2;
            break;
        case 3:
            b = a + 3;
            break;
        default:
            b = a + 4;
            break;
    }
    return b;
}
``` | [ 1, 2, 3, 4, 5 ] | [ 2, 4, 6, 8, 9 ] |
| initialize string with char array | ```csharp
string foo() {
    char[] a = {'a', 'b', 'c'};
    string b = new string(a);
    return b;
}
``` | [ ] | [ "abc" ] |
| array traversal | ```csharp
int foo() {
    int[] arr = new int[]{1, 2, 3};
    int b = 0;
    foreach (int a in arr) {
        b += a;
    }
    return b;
}
``` | [ ] | [ 6 ] |

Figure 6: Examples of C# SUT cases.

## B Models for evaluation

- **TransCoder** (Rozière et al., 2020) is an unsupervised program translation model that can translate functions between C++, Java, and Python with high accuracy.

- **TransCoder-ST** (Rozière et al., 2021) enhances TransCoder by leveraging an automated unit-testing system to filter out invalid translations in the back-translation process.

- **text-davinci-003** originated from Instruct-GPT (Ouyang et al., 2022), the OpenAI team continues to use Instruction Learning and Reinforcement Learning from Human Feedback (RLHF) based on the GPT series of models to guide the training of the model.

- **gpt-3.5-turbo** originated from Chat-GPT (OpenAI., 2022), it is similar to the training method of InstructGPT, and adding more high-quality training data, making it be the currently one of the most competitive code generation models.

## C Detailed Syntax Elements Probing

According to the translations and error messages of failed cases, we manually probe the syntactic defects for these models. For simplicity, we analyze one language pair for each model since our methodology can easily be generalized to other languages. Specifically, we focus on the C++→Python translation task for TransCoder/TransCoder-ST, and the Python→C++ translation task for text-davinci-003/gpt-3.5-turbo. The probed syntactic defects for each model are summarized as follows:

**TransCoder**

i. *update expression*: C++ increment and decrement operators ("++","−") have different execution logic when they are used as prefix and postfix. The model is totally confused about its syntax.

ii. *long keyword*: The model naively copies "`long`" to the translation, while Python has no support for this type.

iii. *comma expression*: C++ comma expression is a way to simplify code. It can be used in an assignment of several variables (e.g., "`a,b=1,2`"), or used in the init/loop expression of "`for`" statement to contain multiple

statements. The model sometimes translates it incorrectly.

iv. *do-while statement*: "`do-while`" statement is an exclusive usage to write a loop in C++, and the model keeps it in the translation which causes syntax errors.

v. *switch-case statement*: "`switch-case`" statement is also an exclusive usage to write a branch in C++, and the model translates it into a serial "`try-except`" expression.

vi. *conditional expression*: C++ ternary operator ("?:") is used to write a branch in the simplest way, and one-line "`if-else`" is used in Python for the same syntax. The model cannot understand its correct translation.

**TransCoder-ST**

TransCoder-ST improves on the syntactic defects in TransCoder. However, it still cannot fully understand the defect i, ii, iii above.

**text-davinci-003**

text-davinci-003 can easily solve many fundamental problems that are prone to errors in the Transcoder series model, but it will still be confused by some of the carefully designed problems.

i. *preserved words*: There are many system reserved word in python, c++, c# and java, which are not allowed to be used as variable names, but the reserved word in different languages only partially overlap, which makes the variable names in the source language may be unacceptable in the target language, and models are often copied directly for use.

i. *variable initialization*: Initialization in different languages often has different implementation methods. In some scenarios, it is random initialization, while in others, default values are assigned, which often confuses the current powerful LLMs(large language models).

ii. *data structure operation*: There are some special data types or relatively unpopular operations among them, and the text-davinci-003 model cannot be well understood and supports mapping to corresponding implementation methods in other languages, such as the del method of list in Python.

ii. *dynamic data type*: Python can disregard the type requirements of strongly typed languages

for assignment, but at runtime, it will distinguish the type of value that the variable points to, which is still difficult for static languages such as C++ to implement automatic type derivation.

**gpt-3.5-turbo**

gpt-3.5-turbo can sometimes cleverly identify scenarios that are prone to confusion in text-davinci-003, especially in the i, ii, but it still cannot perfectly translate all types of code correctly, especially when the input code structure is complex.

## D  Syntax Fine-Tuning (S-Tuning)

In this section, we present a solution for improving the weak points of model translation based on our diagnosis report. It is also very convenient to use this report to expand the model's capabilities in other ways.

### D.1  S-Tuning Dataset

Through the analysis report, we found that the model is relatively weak in the mapping ability of the special syntax structure of each language, which is especially reflected in Transcoder and Transcoder-st. We introduce an unsupervised data augmentation method called Syntax Fine-Tuning (S-Tuning) that leverages syntactic defects probed on our SUT dataset to generate syntactically correct parallel samples for supervised fine-tuning. All syntactic structures in a source language which are not present in a target language can be transformed to equivalent but more elementary structures which are closer in appearance o structures in the target language. For example, when translating from C++ to Python, `int x=i++;` can be transformed into `int x=i; i=i+1;`, which is much closer to a correct Python implementation. This can be viewed as analogous to classical syntax-tree transformation transcompilers, but is a heuristic to bring the source and target closer together to improve the ability of statistical machine learning models to learn to map patterns. With the set of automatic syntax transformations in hand, we augment the existing training data, which we call the S-Tuning dataset, to modify the source language code snippets and fine-tune any existing model to improve its ability to translate elementary syntactic structures. The S-Tuning dataset is built based on a large-scale monolingual dataset of each source language under consideration, we show some case in Fig. 8 For

| Step | Code |
|---|---|
| Source C++ | ```int foo(int a) {    int b = ++a;    return b; }``` |
| Logically equivalent modification | ```int foo(int a) {    a += 1;    int b = a;    return b; }``` |
| Translation of the modification | ```def foo ( a ) :    a += 1    b = a    return b``` |

Figure 7: An example of creating a S-Tuning sample of C++ prefix increment for C++→Python translation. The source C++ function and translation of the modification are combined as a parallel sample.

each syntax element, the generation of its S-Tuning dataset consists of three steps:

1. **Modification**: For each source language program $x$ that contains the syntax element, we generate a logically equivalent function $x'$ with syntax as close as possible to a syntax of the target language.

2. **Translation**: Assuming the model will understand the commonly used syntax structures, we use an existing trained transcompilation model like TransCoder to translate the modified function $x'$ into the target language program $y$.

3. **Reinforcement**: We then fine-tune the transcompilation model like TransCoder to translate source language program $x$ into the translated target language program $y$, helping the model associate the previously poorly understood syntax elements with their proper translation in the target language.

Let us take an example to illustrate the process of S-Tuning dataset generation. The TransCoder-ST (Rozière et al., 2021) model achieves strong performance on translation among C++, Java, and Python. However, it cannot always correctly translate some basic expressions such as prefix increment operation. "++" operator has subtle syntax in C++. The prefix and postfix "++" have different semantics depending on the program context. The process of creating the S-Tuning dataset for prefix increment is shown in Fig. 7.

Table 3: Statistics of the S-Tuning datasets in our experiment. Note that TransCoder/TransCoder-ST requires monolingual data for fine-tuning.

| Model | Parallel | Monolingual | |
|---|---|---|---|
| | # source-target | # source | # target |
| TransCoder | 120k | 200k | 200k |
| TransCoder-ST | 60k | 200k | 200k |

## D.2    S-Tuning Experiment

### D.2.1    S-Tuning Dataset Generation

IBM Project CodeNet[5] dataset is used as our monolingual data source. It is a large-scale curated dataset with diverse implementations where those defects can be easily matched. For each defect, we first search in the monolingual dataset to match the function $x$ that contains the syntax. Then, the function is transformed into a logically equivalent form $x'$ that is familiar to the model based on the modification rules shown in Fig. 8. Next, the vanilla model is used to generate the correctly translated function $y$. Finally, the original function $x$ and the translated function $y$ are paired as a parallel sample in the S-Tuning dataset. During the S-Tuning of TransCoder/TransCoder-ST, we note that fine-tuning solely with parallel functions has negative effects on model performance, and other training steps with monolingual functions are also indispensable.

### D.2.2    Fine-tuning Details

**Dataset**    For TransCoder/TransCoder-ST, we match 20,000 C++ functions for each defect to generate parallel S-Tuning dataset, and we also extract 200,000 C++ and 200,000 Python functions as monolingual datasets for other training steps in vanilla TransCoder (i.e., masked language modeling, denoising auto-encoding, back-translation). The statistics of the S-Tuning datasets in our experiment are shown in Table 3.

**Training**    We initialize the three models with their best checkpoints released on GitHub. The models are optimized with the Adam (Kingma and Ba, 2015) optimizer, a learning rate of $2 \times 10^{-5}$. Other settings are the same as the original. We fine-tune the models for a maximum of 100 epochs, and save the model parameters in every epoch. The final models are selected mainly based on the SUT accuracy. The fine-tuning experiments are conducted

[5]https://github.com/IBM/Project_CodeNet

Table 4:    S-Tuning results of TransCoder and TransCoder-ST C++→Python models.

| Model | TransCoder GeeksforGeeks | | | | SUT |
|---|---|---|---|---|---|
| | EM | BLEU | CodeBLEU | CA@1 | Acc |
| TransCoder | 9.6 | 68.8 | 69.9 | 47.1 | 48.0 |
| TransCoder (S-Tuning) | **9.7** | **69.1** | **70.4** | **50.6** | **63.9** |
| TransCoder-ST | 9.8 | 70.4 | 70.1 | 61.3 | 52.0 |
| TransCoder-ST (S-Tuning) | **9.9** | **70.6** | **70.5** | **63.4** | **70.4** |

in one NVIDIA Tesla V100 GPU.

## D.3    S-Tuning Results and Discussion

Table 4 shows the S-Tuning results of TransCoder and TransCoder-ST C++→Python models. After S-Tuning, TransCoder improves with 15.9% SUT accuracy and 3.5% CA@1, and TransCoder-ST improves with 18.4% SUT accuracy and 2.1% CA@1. We believe that if we fine-tune the models for more defects, the model performance can be further enhanced. We note that static metrics (EM, BLEU, CodeBLEU) have marginal improvements after S-Tuning, which is consistent with our motivation that these metrics cannot assess the syntactic correctness via program execution. Programs with subtle literal differences will have high EM/BLEU/-CodeBLEU scores while they could lead to completely different computation results.

Moreover, we observe that our S-Tuning procedures cannot be easily applied to some syntax elements. For example, template functions in C++ are highly flexible and hard to use rules to describe logically equivalent modifications. Therefore, we skip such syntactic defects even though we have probed them. How to patch those highly flexible syntactic defects remains an intractable task that is worth exploring in the future work.

| Language | Syntactic Defect | Example | Logically Equivalent Modification |
|---|---|---|---|
| C++→Python | i. *update expression* | ```cpp\nint foo(int a) {\n    int b = ++a;\n    return b;\n}\n``` | ```cpp\nint foo(int a) {\n    a += 1;\n    int b = a;\n    return b;\n}\n``` |
| | ii. *long keyword* | ```cpp\nlong foo() {\n    long a(1);\n    return a;\n}\n``` | ```cpp\nint foo() {\n    int a(1);\n    return a;\n}\n``` |
| | iii. *comma expression* | ```cpp\nint foo() {\n    int i, j;\n    for (i = 0, j = 0;\n        i < 5; i++) {\n        j += i;\n    }\n    return j;\n}\n``` | ```cpp\nint foo() {\n    int i, j;\n    i = 0;\n    j = 0;\n    for (; i < 5; i++) {\n        j += i;\n    }\n    return j;\n}\n``` |
| | iv. *do-while statement* | ```cpp\nint foo() {\n    int a = 0;\n    do {\n        a = a + 1;\n    } while (a < 5);\n    return a;\n}\n``` | ```cpp\nint foo() {\n    int a = 0;\n    a = a + 1;\n    while (a < 5) {\n        a = a + 1;\n    }\n    return a;\n}\n``` |
| | v. *switch-case statement* | ```cpp\nint foo() {\n    int a = 0, b = 0;\n    switch (a) {\n        case 0:\n            b = 1; break;\n        case 1:\n            b = 2; break;\n        default:\n            b = 3;\n    }\n    return b;\n}\n``` | ```cpp\nint foo() {\n    int a = 0, b = 0;\n    if (a == 0) {\n        b = 1;\n    }\n    else if (a == 1) {\n        b = 2;\n    }\n    else {\n        b = 3;\n    }\n    return b;\n}\n``` |
| | vi. *conditional expression* | ```cpp\nint foo() {\n    int a = 1;\n    int b;\n    b = (a > 0) ? 1 : -1;\n    return b;\n}\n``` | ```cpp\nint foo() {\n    int a = 1;\n    int b;\n    if (a > 0) {\n        b = 1;\n    } else {\n        b = -1;\n    }\n    return b;\n}\n``` |

Figure 8: The manually probed syntactic defects for the models studied. For each defect, we show a code snippet example and its corresponding logically equivalent modification.