# OpenReview forum: "SUT: Active Defects Probing for Transcompiler Models"
_EMNLP/2023/Conference — EMNLP 2023 Main_

### Official Review · Reviewer_ZEWW · 2023-08-02

**Soundness:** 4

**Excitement:**

3: Ambivalent: It has merits (e.g., it reports state-of-the-art results, the idea is nice), but there are key weaknesses (e.g., it describes incremental work), and it can significantly benefit from another round of revision. However, I won't object to accepting it if my co-reviewers champion it.

**Missing References:**

- Marco Tulio Ribeiro, Tongshuang Wu, Carlos Guestrin, and Sameer Singh. 2020. Beyond Accuracy: Behavioral Testing of NLP Models with CheckList. In Proceedings of the 58th Annual Meeting of the Association for Computational Linguistics, pages 4902–4912, Online. Association for Computational Linguistics.
- Guanqun Yang, Mirazul Haque, Qiaochu Song, Wei Yang, and Xueqing Liu. 2022. TestAug: A Framework for Augmenting Capability-based NLP Tests. In Proceedings of the 29th International Conference on Computational Linguistics, pages 3480–3495, Gyeongju, Republic of Korea. International Committee on Computational Linguistics.
- Aakanksha Naik, Abhilasha Ravichander, Norman Sadeh, Carolyn Rose, and Graham Neubig. 2018. Stress Test Evaluation for Natural Language Inference. In Proceedings of the 27th International Conference on Computational Linguistics, pages 2340–2353, Santa Fe, New Mexico, USA. Association for Computational Linguistics.
- Marco Tulio Ribeiro and Scott Lundberg. 2022. Adaptive Testing and Debugging of NLP Models. In Proceedings of the 60th Annual Meeting of the Association for Computational Linguistics (Volume 1: Long Papers), pages 3253–3267, Dublin, Ireland. Association for Computational Linguistics.

**Paper Topic And Main Contributions:**

The authors propose a new test pipeline and two metrics to improve the testing of program translation models. The new test pipeline has no complicated dependencies and is easier to configure. The authors collect a test suite of 4 programming languages to test their method. The experimental results show that the proposed approach could incur more model failures for both open-source and closed-source models (i.e., `text-davinci-003` and `gpt-3.5-turbo`).

**Questions For The Authors:**

1. The authors try to collect their test suites in a top-down approach, i.e., the test cases are organized by different categories. However, I could only find the aggregate accuracy (Table 2). Could authors also show the per-category error rate?

2. Some numbers in Table 2 are wrongly computed (i.e., 59.64% and 66.41%). I could not get these numbers by either averaging the first 6 columns or 8 columns.

**Reasons To Accept:**

- The proposed approach is technically sound and simplifies the processing of testing code generators because of Python unit tests.
- The idea to collect test suites in a top-down approach (i.e., first defining test categories and then collecting test cases) is novel in the code domain. This practice is reminiscent of a series of behavioral testing works in NLP models (for example, "Beyond Accuracy: Behavioral Testing of NLP models with CheckList"), which also try to organize test cases by specifications.
- The idea of decomposing a unit test into syntax elements and computing the element failure rate (Section 3.2.2) is novel and inspiring. The NLP community could benefit from a similar approach when testing models to generate non-code texts.

**Reasons To Reject:**

- The motivation for this work is not strong enough. Specifically
   - The authors do not clearly explain how translating programs from the source language to the target language could be useful. If the program translation is not a real problem, how could improving the testing of such models be useful?
  - The authors claim that previous metrics like BLEU, ROUGE, and CodeBLEU are not good enough but do not provide detailed explanations or citations (L050-L056).

- The authors do not provide a detailed validity analysis of the test suites (L153-157). Specifically, are these test cases validated to be correct by held-out annotators?


**Reproducibility:**

4: Could mostly reproduce the results, but there may be some variation because of sample variance or minor variations in their interpretation of the protocol or method.

**Reviewer Confidence:**

5: Positive that my evaluation is correct. I read the paper very carefully and I am very familiar with related work.

---

> ### Author Rebuttal · Authors · 2023-08-28
>
> Thank you time and energy to review our manuscript! We read your opinions carefully and we want to explain further about these points of opinions.
>
>  -   The authors do not clearly explain how translating programs from the source language to the target language could be useful. If the program translation is not a real problem, how could improving the testing of such models be useful?
>
>  I don't quite understand your concerns here. My understanding of your question is as follows, you probably don't quite understand how our evaluation tool can improve the performance of the model. Here, you can refer to Appendix D of our paper, Syntax Fine Tuning (S-Tuning) section, which is actually tailored to the situation. For open-source models such as Transcoder, we can mine similar samples with high error rates in our current evaluation samples as training data. After fine-tuning, we can improve the model's performance in this area.
>
> -  The authors claim that previous metrics like BLEU, ROUGE, and CodeBLEU are not good enough but do not provide detailed explanations or citations (L050-L056).
>
> Sorry, this was our oversight. The shortcomings of these n-gram statistical indicators (BLEU, ROUGE, and CodeBLEU) have been analyzed in Transcoder's paper. Here are the original words of the paper,
> "A limitation of  these metrics is that they do not take into account the syntactic correctness of the generations. Two programs with small syntactic discrepancies will have a high BLEU score while they could lead to
> very different compilation and computation outputs. Conversely, semantically equivalent programs
> with different implementations will have low BLEU scores."
> We will add it in the official version.
>
> - The authors try to collect their test suites in a top-down approach, i.e., the test cases are organized by different categories. However, I could only find the aggregate accuracy (Table 2). Could authors also show the per-category error rate?
>  Sure. Here are the per-category error rate
>
> | |Variable&Constant |Datatype|Data Structure|Operator|Loop&Branch|
> |------|------|------|------|------|------|
> |TransCoder|27.09%|12.31%|44.33%|7.3%|8.86%|
> |TransCoder-ST|22.17%|12.35%|46.3%|6.17%|7.41%|
> |text-davinci-003|49.3%|7.04%|28.17%|7.04%|8.45%|
> |gpt-3.5-turbo|55.56%|6.67%|22.22%|6.67%|8.89%|
>
> - Some numbers in Table 2 are wrongly computed (i.e., 59.64% and 66.41%). I could not get these numbers by either averaging the first 6 columns or 8 columns.
>
> Let me calculate here.
>
> The average performance of the text-davinci-003 model on SUT Acc is calculated as follows:,
> 55.99%+65.31%+58.89%+63.54%+65.60%+41.05%+58.08%+68.67%=477.13%,
> 477.13%/8=59.64125%,
>  taking the first two places is 59.64%
>
> The average performance of the gpt-3.5-turbo model on SUT Acc is calculated as follows:,
> 60.62% +70.45% +62.50% +67.18% +78.99% +50.31% +63.53% +77.70%=531.28%,
> 531.28%/8=66.41%,
>  taking the first two places is 66.41%

---

### Official Review · Reviewer_aNwM · 2023-08-03

**Soundness:** 3

**Excitement:**

4: Strong: This paper deepens the understanding of some phenomenon or lowers the barriers to an existing research direction.

**Missing References:**

- In line 045: “Existing datasets are often criticized for their lack of granularity”, Could you please provide relevant references?(The papers criticized the lack of granularity of the previous datasets)

**Paper Topic And Main Contributions:**

**Summary:**
The paper observes that the current transcompiler (code translation) models still make different elementary syntax mistakes, and the current metrics like BLEU and CodeBLEU may not expose that in an interpretable way. To deal with this issue, the paper proposes a new set of unit tests and metrics to investigate the model abilities in translating the elementary syntax element.

**Main contributions:**
- The paper release a set of unit test in different programming languages (including C++, Java, Python, and C#). The unit tests are categorized into different syntax groups and can be used to have a fine-grained evaluation at the elementary syntax level.
- The paper proposes Syntax Unit Test Accuracy (SUT Acc) and Syntax Element Test Score (SETS), and claims that the proposed metrics are more accurate and in line with human judgment.
- The paper evaluates a set of transcompiler models and shows the issues of these models in dealing with different syntactic element categories.

**Questions For The Authors:**

- How can we scale the proposed method? (How can we automatically generate the data for the target programming language?)
- In Lines 187-188: “as some syntax elements are not by themselves able to render a valid program.” It would be good to add a few examples of these syntax elements (From your dataset items)
- In Lines 205-206: “We then use Lasso regression to get consistent results for the estimated log fail.” Could you please elaborate on why using Lasso regressions and provide a relevant reference(s) on that?
- In Lines 219-221: “The test data in TransCoder still remains some data impurity(mainly on the input of the test sample and the mismatch of the given translation pair logic).” Could you please provide evidence for that? E.g., reference(s) or example(s).
- In Fig 2, what the different colors of the “Syntax Unit Test Table” refers to?
- In Table 2: “To X” and “From X” are the averages over different target program languages? Please specify that in the paper. “Average performance” was a bit unclear.

**Reasons To Accept:**

- The paper provides a set of test units that can be used to evaluate the transcompiler models and diagnose the issues of these models at the syntax element level.
- The paper proposes two metrics to evaluate the models' abilities in translating the syntactic element in different programming languages.
- The provided test units and the proposed metrics were used to evaluate the state-of-the-art models.

**Reasons To Reject:**

- There are some issues in the presentations of the paper. (E.g., Fig 2 and Table 2, please refer to “Questions For The Authors”)
- It is not clear how exactly the dataset was constructed and how it can be extended to a new programming language.
- Some of the evaluation details are not clear. E.g., how were the text-davinci-003 and gpt-3.5-turbo where prompted?

**Reproducibility:**

2: Would be hard pressed to reproduce the results. The contribution depends on data that are simply not available outside the author's institution or consortium; not enough details are provided.

**Reviewer Confidence:**

3: Pretty sure, but there's a chance I missed something. Although I have a good feel for this area in general, I did not carefully check the paper's details, e.g., the math, experimental design, or novelty.

**Typos Grammar Style And Presentation Improvements:**

- Line 081: “We propose a more accurate reliable” -> “We propose a more accurate and reliable” ?

---

> ### Author Rebuttal · Authors · 2023-08-28
>
> Thank you time and energy to review our manuscript! We read your opinions carefully and we want to explain further about these points of opinions.
> -   How can we scale the proposed method? (How can we automatically generate the data for the target programming language?)
>
> The basic syntax element test cases are manually annotated by expert programmers and successfully compiled and run. The dataset we propose is an evaluation dataset, so we value the evaluation quality of the model more than the scale. Subsequently, more types of test cases can be collected and cleaned by crawling programming learning websites to expand the dataset. At present, these websites often have inspection type tags, which are convenient for us to label specific unit tests when further processing.
>
> - Some of the evaluation details are not clear. E.g., how were the text-davinci-003 and gpt-3.5-turbo where prompted?
>
> Sorry, we forgot to mention in the paper, the prompt template we used is
>  > // Convert this from X code to Y code
>
>  > // X code
>
>  > %s
>
>  > // Y code
>
>  X and Y respectively represent the source code language and target code language in the translation task, such as (Java, Python, C++, C #), %s represents the code ontology of the source code language.
>
> -   In Lines 187-188: “as some syntax elements are not by themselves able to render a valid program.” It would be good to add a few examples of these syntax elements (From your dataset items)
>
> Sure. Here is the example.
> ```
> int func(int x)
> {
> 	int i;
> 	for(i=0;;++i){
> 		if (x^i=0){
> 		    return ++i
> 		}
> 	return 0
> }
> ```
> Although we have made every effort to construct simple and independent unit tests for each syntactic element, some cases still struggle to achieve true decoupling and merging. For the loop and branch parts, their judgment conditions partially rely on logical expressions, while the block part may also have variable assignments, operator operations, and other parts. When such a case makes an error, it is actually difficult to define which part of it went wrong.  So these examples will be repeatedly labeled under different categories, and further decoupling operations (such as SET scores) need to be relied on to evaluate their more detailed differences under different categories.
>
>
> - In Lines 205-206: “We then use Lasso regression to get consistent results for the estimated log fail.” Could you please elaborate on why using Lasso regressions and provide a relevant reference(s) on that?
>
> Given that we explained in the previous question that even though we have made every effort to build code for relatively independent modules, there are still unit tests that are coupled together. We use Lasso regression here to make the regression coefficients more sparse and further decouple the final coupled unit test, making it easier to locate problematic grammar parts.
>
> -   In Lines 219-221: “The test data in TransCoder still remains some data impurity(mainly on the input of the test sample and the mismatch of the given translation pair logic).” Could you please provide evidence for that? E.g., reference(s) or example(s).
>
> Sure. We evaluate the transcoder CA data from the official code library, [CodeGen/data/transcoder_evaluation_gfg at main · facebookresearch/CodeGen (github.com)]
>
> For example, this question SMALLEST_OF_THREE_INTEGERS_WITHOUT_COMPARISON_OPERATORS_1
> the java code
> ```
> static int f_gold ( int x , int y , int z ) {
>   if ( ( y / x ) != 1 ) return ( ( y / z ) != 1 ) ? y : z ;
>   return ( ( x / z ) != 1 ) ? x : z ;
> }
> ```
> the c++ code
> ```
> int f_gold ( int x, int y, int z ) {
>   if ( ! ( y / x ) ) return ( ! ( y / z ) ) ? y : z;
>   return ( ! ( x / z ) ) ? x : z;
> }
> ```
> ( y / x ) != 1 in Java not equal  ! ( y / x ) in C++
>
> Their logic does not match, which results in the model not passing the case even if it is translated correctly.
>
> -   In Fig 2, what the different colors of the “Syntax Unit Test Table” refers to?
>
> In  Fig 2, our ablation experiment explored the error distribution of these tested models. The meanings represented by these colors are displayed in the upper right corner of the table, where we categorize errors into five categories for specific analysis. (Variable&Constant corresponds to purple, Datatype corresponds to orange, Data Structure corresponds to green, Operator corresponds to red, and Loop &Branch corresponds to purple.), There are still more detailed categories under these five categories, which are marked in the name column of the table. These more detailed error analyses are detailed in Appendix C.
>
>  - In Table 2: “To X” and “From X” are the averages over different target program languages? Please specify that in the paper. “Average performance” was a bit unclear.
>
> Due to space limitations in the short article, we converted the subtasks in the experiment to this format to accommodate the submission template. Actually we have briefly explained the meanings of To X and To Y in Table 2 of the paper, "To X"&"From X": average performance when translating to&from language ". Here, let's give an example to explain in detail that our current evaluation method supports four languages: Python, C#, C++, and Java. So, To Python represents the average pass rate of the three translation tasks: C# to Python, C++ to Python, and Java to Python. From Python represents Python to C #, Python to C++, and Python to Java. Average performance refers to the average of all To X and From X effects at the front of the table.

---

### Official Review · Reviewer_cbZo · 2023-08-03

**Soundness:** 1

**Excitement:**

1: Poor: I cannot identify the contributions of this paper, or I believe the claims are not sufficiently backed up by evidence. I would fight to have it rejected.

**Paper Topic And Main Contributions:**

The paper presents testing approach for programming code made by neural translator. The experiment seems interesting to me as a computer scientist but I fail to see the relation of the proposal to natural language processing which is the main topic of the conference.

**Reasons To Accept:**

* probably the experiments may interest some software engineers at emnlp

**Reasons To Reject:**

* programming languages are not natural languages
* there's not much content related to conference topic at all and paper makes no explanations to bridge the gap

**Reproducibility:**

3: Could reproduce the results with some difficulty. The settings of parameters are underspecified or subjectively determined; the training/evaluation data are not widely available.

**Reviewer Confidence:**

1: Not my area, or paper was hard for me to understand. My evaluation is just an educated guess.

---

> ### Author Rebuttal · Authors · 2023-08-28
>
> Thank you time and energy to review our manuscript! We read your opinions carefully and we want to explain further about these points of opinions.
>
> The research objectives of natural language processing not only include languages that have evolved with culture, such as English and French, but also various languages created by humans specifically for certain purposes, such as Esperanto, code languages, tables, etc. Their common feature is that they both contain rich information and are tools used by people to communicate and collide with each other's thinking.
>
> The following are relevant papers on programming languages (or code languages) that have been published in top-level natural language processing conferences such as ACL and EMNLP in recent years, for your reference, thank you!
>
> **ACL2023:**
> - **[ReCode: Robustness Evaluation of Code Generation Models]**
> - **[Self-Edit: Fault-Aware Code Editor for Code Generation]**
> - **[Code4Struct: Code Generation for Few-Shot Event Structure Prediction]**
> - **[Multi-target Backdoor Attacks for Code Pre-trained Models]**
> - **[DIP: Dead code Insertion based Black-box Attack for Programming Language Model]**
> - **[Backdooring Neural Code Search]**
>
> **EMNLP2022:**
> - **[CoCoa: An Encoder-Decoder Model for Controllable Code-switched Generation]**
> - **[CodeRetriever: A Large Scale Contrastive Pre-Training Method for Code Search]**
> - **[Exploring Representation-level Augmentation for Code Search]**
> - **[Towards Compositional Generalization in Code Search]**
> - **[Language Models of Code are Few-Shot Commonsense Learners]**
>
> Our paper, as well as the aforementioned papers, both discuss tasks related to understanding and generating code. We present a new dataset and evaluation scheme in our paper, which can objectively evaluate the performance of current large-scale language models in code translation tasks. The experiments in our main text and appendix also explore how to use our evaluation tools to improve the performance of these language models. Please review our paper again based on our rebuttal. Thank you!

---

### Meta-Review · Area_Chair_ZoJm · 2023-09-14

**Recommendation:** 4

**Metareview:**

The main conclusions of the reviews and the post-rebuttal discussions:

2/ 3 reviewers consider the paper sound (scores 1, 3, 4)
2/ 3 reviewers find the paper exciting (scores 1, 4, 3)
From reading the rebuttal and seeing the scores above, I find that the reviewers consider strong points for soundness the following:

- The paper observes that the current transcompiler (code translation) models still make different elementary syntax mistakes, and the current metrics like BLEU and CodeBLEU may not expose that in an interpretable way.
- To deal with this issue, the paper proposes a new set of unit tests and metrics to investigate the model abilities in translating the elementary syntax element.
- The idea of decomposing a unit test into syntax elements and computing the element failure rate is novel and inspiring. The NLP community could benefit from a similar approach when testing models to generate non-code texts.

Reviewers had questions regarding dataset construction, evaluation, and how it could be extended to a new programming language. The authors answered these questions in the rebuttal and the reviewers accepted their answers and gave constructive feedback for improving the paper.

---

### Decision · Program_Chairs · 2023-10-07

**Decision:**

Accept-Main

**Comment:**

The main conclusions of the reviews and the post-rebuttal discussions:

2/ 3 reviewers consider the paper sound (scores 1, 3, 4)
2/ 3 reviewers find the paper exciting (scores 1, 4, 3)
From reading the rebuttal and seeing the scores above, I find that the reviewers consider strong points for soundness the following:

- The paper observes that the current transcompiler (code translation) models still make different elementary syntax mistakes, and the current metrics like BLEU and CodeBLEU may not expose that in an interpretable way.
- To deal with this issue, the paper proposes a new set of unit tests and metrics to investigate the model abilities in translating the elementary syntax element.
- The idea of decomposing a unit test into syntax elements and computing the element failure rate is novel and inspiring. The NLP community could benefit from a similar approach when testing models to generate non-code texts.

Reviewers had questions regarding dataset construction, evaluation, and how it could be extended to a new programming language. The authors answered these questions in the rebuttal and the reviewers accepted their answers and gave constructive feedback for improving the paper.